# Neural Superposition Networks

## Abstract

Machine learning models can be biased towards the solutions of given differential equations in two principal ways: through regularisation, or through architecture design. Recent research has successfully constrained neural network architectures to satisfy divergence-free fields and Laplace's equation in two dimensions. This work reinterprets these architectures as linear superpositions of general formulated solutions. The notion of superposition is then exploited to develop novel architectures which satisfy both these and novel differential equations. In addition to new architectures for Laplace's equation and divergence-free fields, we propose novel constraints apt for the heat equation, and even some nonlinear differential equations including Burgers' equation. Benchmarks of superposition-based approaches against previously published architectures and physics-informed regularisation approaches are presented. We find that embedding differential equation constraints directly into neural network architectures can lead to improved performance and hope our results motivate further development of neural networks architectures developed to adhere specifically to given differential constraints.

## 1 Introduction

There has been fruitful interplay between the fields of machine learning and differential equation solving. Differential equations provided useful means to machine learning: ordinary differential equations (ODEs) appear in continuous depth neural networks (Chen et al., 2018) and stochastic differential equations (SDEs) inspired novel approaches to generative modelling (Song et al., 2020). On the other side, neural network architectures have also provided novel means to analyse and solve differential equations, in particular, physics-informed neural networks (PINNs) (Lagaris et al., 1998; Raissi et al., 2019) bias neural network outputs towards a given differential equation via construction of appropriate regularisation terms. Also, neural networks have been shown to be effective surrogate models of computationally-expensive, finite element solvers of partial differential equations (PDE) (Nabian & Meidani, 2018). In addition to providing novel means for solving forward problems of differential equations, neural networks proved useful in solving inverse-problems in differential equations (Lu et al., 2021b), data-driven discovery of differential equations (Both et al., 2021) and approaches for optimal control (Mowlavi & Nabi, 2023).

Thus far, physics-informed regularisation has been a dominant approach to biasing machine learning models towards differential equations, being widely applied from PDEs and ODEs (Raissi et al., 2019) to fractional order differential equations (Pang et al., 2019). Despite their widespread applicability, the difficulty of training PINNs has been highlighted in many works that also highlight potential means to improve their training. For example, via appropriate loss-scaling techniques (Wang et al., 2021; Son et al., 2023), domain decomposition (Jagtap & Karniadakis, 2020; Moseley et al., 2023), combining gradient and Hessian-based optimisation procedures (Mao et al., 2020), and relaxation of the underlying differential equation form (Krishnapriyan et al., 2021).

It is yet unclear how to reliably train PINNs, or eventually provide convergence guarantees. Therefore, recent research has started to move beyond PINNs, towards embedding target differential equations into underlying architectures directly. For example, neural conservation laws (NCLs), which involve post-processing of multilayer perceptrons (MLPs) with higher-order derivatives to yield divergence-free neural networks, ensure conservation principles by using divergence-free vector fields (Richter-Powell et al., 2022). Holomorphic neural networks, which are complex-valued MLPs with appropriate activation functions, have been shown to satisfy the Laplace equation (Ghosh et al., 2023). Additionally, neural networks satisfying Hamiltonian laws have been developed to preserve

the total energy of a dynamical system over time (Greydanus et al., 2019). Beyond the context of neural networks, priors for Gaussian processes over the space of solutions of given linear differential equations have also been developed (Harkonen et al., 2023).

**Contributions of this work** This work presents a framework for constructing neural networks to satisfy linear differential equations through the principle of superposition. Since the solution space of homogeneous linear differential equations is closed under addition, we can derive single-layer feedforward architectures that leverage a superposition of solutions as a learnable representation of solution functions. We demonstrate how this framework provides a unifying perspective on previously published neural network architectures applying to divergence-free vector fields (Richter-Powell et al., 2022) and Laplace's equations (Ghosh et al., 2023). Novel architectures expanding on such applications to encompass also the heat equation and Burgers' equation are presented. Despite Burgers' equation being nonlinear, we are able to construct appropriate architectures constrained to it via suitable post-processing of architectures satisfying the heat equation. We benchmark our architectures against representative, standard architectures constrained to the same differential equations (where applicable) as well as physics-informed neural networks.

## 2 THEORY

We consider neural network architectures constrained to satisfy homogeneous linear differential equations.

Consider an open set $\Omega \subset \mathbb{R}^{d_1}$, $d_1 \in \mathbb{N}$ with boundary $\partial\Omega$. Define a space of sufficiently smooth functions, $\mathcal{A}$, from $\Omega \cup \partial\Omega$ to $\mathbb{R}^{d_2}$, $d_2 \in \mathbb{N}$. We consider equations of the form

$$\begin{aligned} \mathcal{L}u &= 0 \text{ in } \Omega \\ \mathcal{N}u &= 0 \text{ on } \partial\Omega \end{aligned} \tag{1}$$

where $u \in \mathcal{A}$, $\mathcal{L}$ and $\mathcal{N}$ are linear differential operators mapping elements of $\mathcal{A}$ to $\mathbb{R}^{d_3}$-valued functions on $\Omega \cup \partial\Omega$, i.e. that $\mathcal{L}(af + bg) = a\mathcal{L}f + b\mathcal{L}g$ for all $f, g \in \mathcal{A}$ and $a, b \in \mathbb{R}$. In the case where $d_3 > 1$ is vector-valued, we take the right-hand side to be the zero-vector.

Note that the assumptions in our formulations encompass a wide-range of common differential equations with wide-reaching practical applications, such as the Laplace, Diffusion, Heat, Sturm-Liouville and Wave equations, to name but a few. In all these cases, the goal is to find a function $u$ satisfying the differential equation and boundary conditions imposed by respectively $\mathcal{L}$ and $\mathcal{N}$ in Eq. equation 1.

Define a family of parameterised functions, $u_{\theta_i}^i$, $i = 1, \ldots, n$ for $n \in \mathbb{N}$, with $u_{\theta_i}^i : \mathbb{R}^{d_1} \to \mathbb{R}^{d_2}$ and $\theta_i$ representing trainable parameters. Note that superscripts denote positional elements of $\mathbf{u}$; they do not represent powers. Interpreting each $u_{\theta_i}^i$ to be a row vector comprising of $d_2$ columns, we can represent the family of functions $u_{\theta_i}^i$ as an $n \times d_2$ matrix:

$$\mathbf{u} = \begin{pmatrix} u_{\theta_1}^{1(1)} & \cdots & u_{\theta_1}^{1(d_2)} \\ \vdots & \ddots & \vdots \\ u_{\theta_n}^{n(1)} & \cdots & u_{\theta_n}^{n(d_2)} \end{pmatrix} \tag{2}$$

we allow $\mathbf{u}$ to be interpreted as a matrix-valued function operating on $\Omega \cup \partial\Omega$, where the $\mathbf{u}(x)$ is determined by the element-wise application of each element of $\mathbf{u}$ to $x$.

**Definition 1 (Superposition network)** *Given an $1 \times n$ matrix $W$, a function $\phi : (\Omega \cup \partial\Omega) \to \mathbb{R}^{d_2}$, given by $\phi_\theta = W\mathbf{u}$, where $\theta = \{W, \theta_1, \theta_2, \ldots, \theta_n\}$ is referred to as a superposition network with respect to $\mathcal{L}$ if and only if $\mathcal{L}u_{\theta_i}^i = 0$ for all $i$.*

Given a superposition network with respect to $\mathcal{L}$ denoted by $\phi_\theta$, it follows directly that $\mathcal{L}\phi_\theta = 0$ by linearity of $\mathcal{L}$. Thus, by solving the following optimisation procedure:

$$\theta^* = \arg\min_\theta \mathbb{E}_{x \sim \mathbb{P}_{\partial\Omega}} \left[ (\mathcal{N}\phi_\theta(x))^2 \right], \tag{3}$$

where the expectation is taken with respect to a probability distribution $\mathbb{P}_{\partial\Omega}$ defined over $\partial\Omega$. The resulting function $\phi_{\theta*}$ then approximates the solution of Eq. equation 1, as exemplified in Fig. 1.

The construction of superposition networks with respect to a given linear-differential operator $\mathcal{L}$ depends on the specific operator at hand. This is exemplified in section 2.1, where we note that existing architectures in the literature constrained to Laplace's equation and divergence-free constraints can be interpreted as superposition networks, and section 2.2 where we propose new architectures for additional operators.

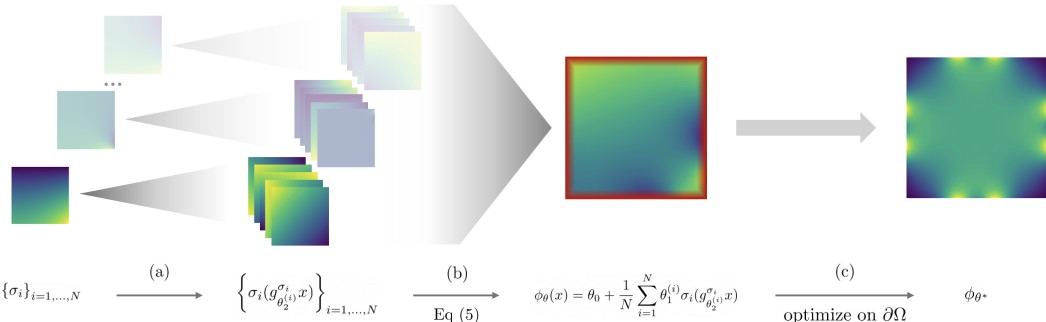

Figure 1: A schematic of superposition networks, a single-layer feedforward neural network architecture constrained to be in the solution space of a linear differential equation. Superposition networks use a library of known solutions of the differential equation (a) and apply Lie group symmetries derived from the differential equation to derive suitable linear transformations (b) which can then be combined in an output layer (c) to approximate nontrivial solutions of the differential equation by training only on initial and boundary conditions.

## 2.1 EXISTING SUPERPOSITION NETWORKS

This section demonstrates that the principle of superposition provides a unifying perspective on previous neural network architectures used to model divergence-free fields (Richter-Powell et al., 2022) ($\mathcal{L} = \nabla\cdot$) and Laplace's equation (Ghosh et al., 2023) ($\mathcal{L} = \nabla^2$). Both of these works apply a postprocessing step to an existing neural network to constrain it to a certain differential equation. To unify our treatment of both architectures we introduce the notion of a linear postprocessing operator.

**Definition 2 (Linear postprocessing operator)** *Consider a function $f : \mathbb{R}^{d_1} \to \mathbb{R}^{d_3}$, with components enumerated by $[f_1, f_2, \ldots, f_{d_3}]$. We equate $f$ with its vector representation. Define an operator $A$ acting on $f$ as a linear postprocessing operator with respect to $\mathcal{L}$ if and only if the following hold:*

1. *The action of $A$ on $f$ is $\mathbb{R}$-linear in the components of $f$, i.e. $A[af_1, af_2, \ldots, af_{d_3}] = aA[f_1, f_2, \ldots, f_{d_3}] \, \forall a \in \mathbb{R}$ and*

2. *$\mathcal{L}(A(f)) = 0$.*

*We use the notation $A[f_1, \ldots, f_{d_3}]$ to denote the application of $A$ to $f$, which we write in a vector notation above to make clear that linearity of the operator acts on output components of $f$ as opposed to the inputs of $f$.*

**Example 1 (Divergence-free neural networks)** *In work on modelling conservation laws, neural models of divergence-free vector-fields were derived by considering the Hodge-star operation on the exterior derivative of a differential form (Richter-Powell et al., 2022). The composition of the Hodge-star operator on the exterior derivative is a linear postprocessing operator with respect to the divergence operator.*

**Example 2 (Holomorphic neural networks)** *Taking the real part of a complex-valued MLP with holomorphic activation functions yields a neural network architecture which satisfies Laplace's*

*equation, $\nabla^2 \phi = 0$ (Ghosh et al., 2023). In this scenario, taking the real-part of the complex output of the MLP forms a linear postprocessing operator with respect to the Laplacian.*

If $f$ is taken as an MLP with a linear output layer, we can write $f = [f_1, f_2, \ldots, f_{d_3}]$ and take each $f_i(x) = \sum_j w_{ij}\sigma_j(x)$, where $\sigma_j(x)$ represents the activation of the $j$-th neuron of the last hidden layer of the MLP with input $x$. Application of the linear postprocessing operator can now yield a degeneracy of ways of representing a target neural network. For example, applying linearity with $i = 1$ yields:

$$A[\sum_j w_{1j}\sigma_1, \ldots, f_{d_3}] = \sum_j |w_{1j}|A[\frac{w_{1j}}{|w_{1j}|}|\sigma_1, \ldots, \frac{1}{|w_{1j}|}f_{d3}], \tag{4}$$

with applications of linearity to other indices giving alternative superposition representations.

Note that each $w_{ij}$ can potentially be complex-valued, as in the case for holomorphic neural networks (Ghosh et al., 2023), whereas the definition of linear postprocessing operators only assumes linearity over real numbers. Consequently, we only apply linearity to the absolute value of $w_{1j}$ in Eq. equation 4.

The right hand side of Eq. equation 4 yields the definition of a superposition network if $\mathcal{L}A[\frac{w_{1j}}{|w_{1j}|}|\sigma_1, \ldots, \frac{1}{|w_{1j}|}f_{d3}] = 0$ for each $j$. For instance, this holds for the neural networks presented in Examples 1-2, which can thus be interpreted as special cases of superposition networks.

## 2.2 NOVEL SUPERPOSITION NETWORKS

The use of linear postprocessing operators in previous works (Ghosh et al., 2023; Richter-Powell et al., 2022) is tailored to specific linear differential equations—they use specific characteristics of the exterior calculus of differential forms, or of holomorphic complex-valued functions, to cater to particular differential equations. These approaches are not applicable to arbitrary linear differential equations that we might consider as in Eq. equation 1. This section outlines an applicable approach to systematically address new differential equations. We leverage the Lie group symmetries admitted by a given differential operator to construct superposition networks applicable to cases beyond the current state-of-art. Means to handle some non-linear differential equations are presented (i.e. the Burgers' equation) in section 2.2.4.

Consider superposition networks of the form $\phi : \mathbb{R}^{d_1} \to \mathbb{R}^{d_2}$, given by:

$$\phi_\theta(x) = \theta_0 + \frac{1}{N}\sum_{i=1}^{N}\theta_1^{(i)}\sigma_i(g_{\theta_2^{(i)}}^{\sigma_i}x) \tag{5}$$

where the output layer of the architecture is explicitly summed and the following terms are introduced: (1) The activation function for the $i$th neuron, $\sigma_i : \mathbb{R}^{d_1} \to \mathbb{R}^{d_2}$, satisfies $\mathcal{L}\sigma_i = 0$ for all $i$. These activation functions can be trivial manufactured solutions for boundary conditions other than those specified in Eq. equation 1. (2) Linear transformations which are not freely chosen, but rather elements of a Lie group $g_{\theta_2^{(i)}}^{\sigma_i}$ acting on $\mathbb{R}^{d_1}$ parameterised by trainable parameters $\theta_2^{(i)}$ such that $\mathcal{L}\sigma_i(g_{\theta_2^{(i)}}^{\sigma_i}x) = 0$ whenever $\mathcal{L}\sigma_i(x) = 0$. (3) Parameters associated with an affine output layer as usual: $\theta_0$ and $\theta_1^{(i)}$.

Note that the inclusion of the parameter $\theta_0 \in \mathbb{R}^{d_2}$ generalises definition 1. However, in all differential equations we consider in this work, $\mathcal{L}$ has first-order derivatives which allows us the flexibility to include $\theta_0$ in our architecture. Taking $\theta = \left\{W, \theta_0, \theta_2^{(1)}, \theta_2^{(2)}, \ldots, \theta_2^{(N)}\right\}$, with $W = \left\{\theta_1^{(1)}, \theta_1^{(2)}, \ldots, \theta_1^{(N)},\right\}$ allows us to the approximate a solution to Eq. equation 1 by solving the optimisation problem in Eq. equation 3.

Therefore, constructing appropriate superposition networks for a given linear-differential operator amounts to choosing suitable forms of $\sigma_i$ and $g_{\theta_2^{(i)}}^{\sigma_i}$. In general, this can be done in a methodical way by enumerating the Lie group symmetries of $\mathcal{L}$ (Gray, 2015). However, as we now demonstrate, for several important classes of practical equations, suitable forms can be constructed by observation and the method of manufactured solutions.

### 2.2.1 LAPLACE'S EQUATION

In two-dimensions, Laplace's equation is defined by taking $\mathcal{L} = \frac{\partial^2}{\partial x^2} + \frac{\partial^2}{\partial y^2}$ in Eq. equation 1. While architectures have previously been designed to satisfy the Laplace equation, we present alternative architectures here as a means of exemplifying the construction of superposition networks.

It follows directly that the following two-parameter Lie groups translate solutions of the Laplace equation (harmonic functions) to other solutions of the Laplace equation:

$$(\hat{x}, \hat{y}) = g_{\theta_D}^D g_{\theta_E}^{E2}(x, y) \tag{6}$$

with (trainable) Lie group parameters being given by $\theta_E \in \mathbb{R}^3$ and $\theta_D \in \mathbb{R}$. The transformations in Eq. equation 6 are a composition of $E2$, the Euclidean group in two-dimensions, with a dilation of the two-dimensional plane given by $g_{\theta_D}^D(x, y) = (\theta_D x, \theta_D y)$.

For any function, $\sigma(x, y)$ such that $\left( \frac{\partial^2}{\partial x^2} + \frac{\partial^2}{\partial y^2} \right) \sigma(x, y) = 0 \; \forall (x, y) \in \Omega$, it is readily verifiable that the Laplacian is invariant under group actions as described in Eq. equation 6, which thus define a suitable form for use as the transformation group in Eq. equation 5.

To define suitable functions $\sigma_i$ in Eq. equation 5, we can take the real part of any holomorphic function $f : \mathbb{C} \to \mathbb{C}$, a choice for $f(z) = f(x + iy)$ of for example $\sin z$, $\sin(\sin z)$, $\sin z^2$ or $\sin^2 z$ would yield for $\sigma_i(x, y)$ the function $\sin x \cosh y$, $\sin(x^2 - y^2) \cosh(2xy)$, $\sin^2 x \cosh^2 y - \cos^2 x \sinh^2 y$ and $\sin(\sin x \cosh y) \cosh(\cos x \sinh y)$, respectively.

Combining these choices of basis functions with linear transformations as defined in Eq. equation 6 with Eq. equation 5 yields a single hidden layer superposition network constrained to Laplace's equation.

**Remark 1 (The universality of superposition networks)** *As the weighted summation of the real parts of holomorphic functions, it follows that the Laplacian superposition network as outlined is the real part of a holomorphic function. Such functions cannot represent arbitrary harmonic functions. In the case of $\Omega$ being a multiply-connected domain, there are harmonic functions that cannot be represented as the real-part of a holomorphic function. There are means to circumvent such restrictions (Ghosh et al., 2023). However, the wider topic of proving the universality of a function class within a space of solutions of arbitrary linear differential operators is, to our knowledge, yet an open area of study. We thus leave investigations of universal superposition networks to future work.*

### 2.2.2 DIVERGENCE-FREE FIELDS

Similarly to the Laplace equation constrained architectures of section 2.2.1, divergence-free fields, given in the scenario where $\mathcal{L}\phi = \nabla \cdot \phi$, obey the same symmetries as defined in Eq. equation 6.

It remains to define suitable activation functions. Previous work on deriving divergence-free neural network architectures (Ghosh et al., 2023; Richter-Powell et al., 2022) has demonstrated this approach.

In two-dimensions, given a differentiable function $f : \mathbb{R}^2 \to \mathbb{R}$, we can derive a divergence-free field $g : \mathbb{R}^2 \to \mathbb{R}^2$ via the following transformation:

$$(\hat{x}, \hat{y}) = \left( \frac{\partial f(x, y)}{\partial y}, -\frac{\partial f(x, y)}{\partial x} \right). \tag{7}$$

The explicit forms used to derive superposition networks in our numerical experiments are enumerated in section 3.4. In three-dimensions, given a differentiable function $f : \mathbb{R}^3 \to \mathbb{R}^3$, we can derive a divergence-free field $g : \mathbb{R}^3 \to \mathbb{R}^3$ using:

$$(\hat{x}, \hat{y}, \hat{z}) = \nabla \times f(x, y, z) \tag{8}$$

### 2.2.3 HEAT EQUATION

We consider architectures constrained to satisfy the 2+1 (two spatial and one temporal) dimensional heat equation, where $\mathcal{L} = \frac{\partial}{\partial t} - \alpha \left( \frac{\partial^2}{\partial x^2} + \frac{\partial^2}{\partial y^2} \right)$ in Eq. equation 1, where $\alpha$ is the thermal diffusivity coefficient.

In contrast to section 2.2.1, which constructed generally applicable Lie group symmetries of the Laplace equation, here we demonstrate the construction of a superposition network for an appropriate Lie group via a manufactured solution of the heat equation.

Consider functions $\sigma_i(x, y, t)$ of the following form:

$$\sigma_i(x, y, t) = e^{-2\alpha t} \sin x \sin y, \tag{9}$$

which provides a manufactured solution to the heat equation suitable as the activation function in Eq. equation 5.

Given the heat-equation and the chosen activation function, it can be validated through direct substitution that the following group action provides a trainable transformation $(x, y, t) \to (\hat{x}, \hat{y}, \hat{t})$ for use as the linear component of Eq. equation 5:

$$
\begin{aligned}
\hat{x}_i &= \theta_{x1i}x + \theta_{x2i} \\
\hat{y}_i &= \theta_{y1i}y + \theta_{y2i} \\
\hat{t}_i &= \frac{\theta_{x1i}^2 + \theta_{y1i}^2}{2}t + \theta_{ti},
\end{aligned}
\tag{10}
$$

whence it follows [1] that $\mathcal{L}\sigma_i(\hat{x}_i, \hat{y}_i, \hat{t}_i) = 0$, and hence constraining the linear layer of a superposition network to follow Eq. equation 10, with an activation function given by Eq. equation 9, yields an architecture automatically satisfying the heat equation.

### 2.2.4 BURGERS' EQUATION

So far, we investigated how the construction of superposition networks can constrain architectures to certain *linear* differential operators, leaving unaddressed the case of nonlinear differential equations. However, the latter can be converted to (and from) linear differential equations through appropriate transformations in certain important cases. Applying such transformations to superposition networks thus extends the utility of these architectures to cases of nonlinear differential equations.

To this end, we consider 1D Burgers' equation:

$$\frac{\partial u}{\partial t} + u\frac{\partial u}{\partial x} = \nu\frac{\partial^2 u}{\partial x^2} \tag{11}$$

If $\nu \neq 0$, we can apply the Cole-Hopf transformation (Cole, 1951; Hopf, 1950): $u(x, t) = -2\nu\frac{\partial}{\partial x}\ln\phi(x, t)$ which yields the heat equation

$$\frac{\partial\phi}{\partial t} = \nu\frac{\partial^2\phi}{\partial x^2} \tag{12}$$

If $\phi(x, t)$ is defined via the means outlined in section 2.2.3 with Eq. equation 5, it follows that $\phi(x, t)$ naturally satisfies the heat equation. We can thus train the superposition network on the boundary and initial conditions inherited from the Burgers' equation and finally apply the inverse transformation, where the derivatives can be calculated readily via automatic differentiation. The resulting architecture is then guaranteed to be constrained to Burgers' equation.

This approach can be generalized to 2D and 3D Burgers' equations

$$\frac{\partial\mathbf{u}}{\partial t} + \mathbf{u}\cdot\nabla\mathbf{u} = \nu\nabla^2\mathbf{u} \tag{13}$$

via the transformation $\mathbf{u} = -2\nu\nabla\ln\phi$.

## 3 APPLICATIONS

### 3.1 METHODS

There are two principal manners in which to provide a differential-equation based inductive bias to a neural network: 1. with appropriate regularisation terms, such as physics-informed neural networks(Raissi et al., 2019). 2. by incorporating differential equation constraints within the architecture itself (Ghosh et al., 2023; Richter-Powell et al., 2022).

---

[1]Note how the partial derivatives of $\mathcal{L}$ are still taken with respect to the original coordinates of $(x, y, t)$.

We seek to benchmark superposition networks against both approaches where possible, but limit ourselves to PINNs where no such hard-constrained architecture exists.

**General Formulation** We consider differential equations as outlined in equation 1, however, we allow $\mathcal{L}$ to also be nonlinear to encompass our handling of Burgers' equation.

Introduce probability distributions of collocation points over the interior of a domain $\mathbb{P}_\Omega$ and the boundary of the domain $\mathbb{P}_{\partial\Omega}$.

**PINN(BI)** PINNs use appropriately constructed regularisation terms to provide inductive biases towards a given differential equation. Many neural network architectures are amenable to physics-informed regularisation, yet MLPs tend to be the most widely used. To optimise a PINN to solve a forward-solution of a given differential, the following loss function is minimized:

$$L_{\text{PINN}}(\theta) = \mathbb{E}_{x \sim \mathbb{P}_{\partial\Omega}} \left[ (\mathcal{N} f_\theta(x))^2 \right] + \mathbb{E}_{x \sim \mathbb{P}_\Omega} \left[ (\mathcal{L} f_\theta(x))^2 \right], \tag{14}$$

where $f_\theta$ represents an MLP with trainable parameters and the two terms represents contributions of the boundary and interior respectively. We denote by PINNB/PINNI methods which provide an extra weight (by a factor of 1000 in our numerical experiments) to the boundary and interior loss terms, respectively.

**RAR** The expectation over $\Omega$ in equation 14 is often done over a fixed distribution. In Lu et al. (2021a), an adaptive approach is introduced whereby points with high PDE residuals are adaptively added to the pool of points to estimate of the second term of equation 14. We name such approach RAR and include it within our benchmarks: every 1000 epochs of training, we sample 1024 new candidate collocation points, and add the 32 with the highest PDE residuals to the pool of collocation points used for each subsequent training epochs.

**AA** Adaptive activation functions were proposed by Jagtap et al. (2020) to improve convergence of PINNs. In particular, they propose that given an MLP $f(x) = W_L(\sigma(W_{L-1}(\ldots(\sigma(W_1 x)))))$, that it be modified to be of the form $f(x) = W_L(an\sigma(W_{L-1}(\ldots(an\sigma(W_1 x)))))$ for a trainable parameter $a$ and fixed hyperparameter $n$. Following the original publication, we initialise $a = 1.0$ and fix $n = 10$ for all our experiments. We also include a combination of this technique with the residual adaptive approach which we denote as **RAR+AA**.

**Holomorphic Neural Networks** We implement holomorphic neural networks as a means to model the Laplace equation, following Ghosh et al. (2023).

Holomorphic neural networks parameterise a holomorphic function with a complex-valued MLP $f_\theta(x + iy) : \mathbb{C} \to \mathbb{C}$. If the activation functions of the MLP are themselves holomorphic, then the real part of the output is guaranteed to satisfy Laplace's equation in two-dimensions with inputs $x$ and $y$. A holomorphic neural network can thus be trained to solve instances of Laplace's equations in two-dimensions by minimising the following loss function.

$$L_{\text{Holomorphic}}(\theta) = \mathbb{E}_{x \sim \mathbb{P}_{\partial\Omega}} \left[ \mathcal{N} \left( \text{Re}(f_\theta(x)) \right) \right], \tag{15}$$

where $\text{Re}(z)$ denotes taking the real part of a complex number $z$ and $f_\theta$ represents a holomorphic neural network. In all numerical experiments, we use $\sin$ as our holomorphic activation function.

**NCL** It is possible to derive divergence-free neural networks in arbitrary dimensions (Richter-Powell et al., 2022). However, in our numerical experiments, we limit ourselves to the two-dimensional case, which allows us to simplify our exposition compared to the more general formulations presented in (Richter-Powell et al., 2022).

Define a neural network $f_\theta : \mathbb{R}^2 \to \mathbb{R}$ as a multilayer perceptron with trainable parameters $\theta$. Then we can postprocess $f_\theta$ as follows to achieve a divergence-free function $g_\theta : \mathbb{R}^2 \to \mathbb{R}^2$ given by applying Eq. equation 7 to $f_\theta$. We optimise divergence-free neural networks as per the loss function in Eq. equation 14.

**Experimental Details** Details common for every numerical experiment are outlined in this subsection. All experiments reported in Table 1 were executed in Python3, making use of NumPy (Harris et al., 2020) and Matplotlib (Hunter, 2007) libraries, with all the neural networks implemented

Table 1: A summary of experimental results (lower is better) comparing superposition networks to alternative architectures imposing differential equation constraints for the methods outlined in section 3.1. Root-mean squared errors (RMSEs) of the final trained solution are shown with standard deviations over 10 random seeds reported. For the heat equation, we report the RMSE at the end of the simulation. Note that Holomorphic neural networks are only applicable to Laplace's equation, and NCL only applies to divergence-free fields.

| | Laplace 1 | Laplace 2 | Heat 1 | Heat 2 | Navier Stokes | Burgers' |
|---|---|---|---|---|---|---|
| Superposition | 0.0067±0.0023 | 0.010±0.0047 | 0.0080±7.3e-5 | 0.00084±0.00018 | 0.11±0.0026 | 0.0030±0.0024 |
| PINN | 0.15±0.0030 | 0.29±0.093 | 0.0085±0.0024 | 0.0027±0.0017 | 0.10±0.00090 | 0.0039±0.0022 |
| PINNB | 0.0063±0.0041 | 0.12±0.066 | 0.063±0.016 | 0.015±0.0025 | 0.085±0.0090 | 0.0049±0.0030 |
| PINNI | 0.56±0.00091 | 0.82±0.067 | 0.080±0.024 | 0.046±4.4e-5 | 0.10±0.00034 | 0.12±0.010 |
| RAR | 0.15±0.0015 | 0.43±0.012 | 0.0085±0.0075 | 0.0026±0.0012 | 0.10±0.00058 | 0.0036±0.00065 |
| AA | 0.19±0.12 | 0.54±0.084 | 0.039±0.015 | 0.016±0.016 | 0.097±0.00086 | 0.0067±0.0039 |
| RAR+AA | 0.20±0.12 | 0.55±0.063 | 0.0070±0.0020 | 0.0053±0.0034 | 0.099±0.00080 | 0.018±0.012 |
| NCL | - | - | - | - | 0.097±0.0015 | - |
| Holomorphic | 0.0029±0.0022 | 0.0033±0.0009 | - | - | - | - |

via PyTorch (Paszke et al., 2019). In the supplementary material we also report an independent implementation of the PINN baseline for the Laplace benchmark using JAX Bradbury et al. (2018), which does not materially differ from the numerical results presented.

Ground truths for the heat equation and Navier-Stokes benchmarks were constructed using FEATools commercial software with the open-source OpenFOAM backend (Weller et al., 1998) for Navier-Stokes, and MATLAB's backslash backend for the heat equation (Amestoy et al., 2000). While the source-code is based on proprietary software, we include CSVs of simulation output for use with our analyses for the purpose of reproducibility along with the source code of all experiments in the supplementary material. We consistently use Kaiming Uniform initialisers (He et al., 2015), optimised for 32000 epochs over full-batches with an Adam optimizer (Kingma & Ba, 2014) with a learning rate (LR) of $10^{-3}$. We use $\tanh$ activations for real-valued neural networks and $\sin$ activations for holomorphic networks. All experiments run using Python 3.10 on two-cores of a Dual AMD Rome 7742 processor with 8GB of RAM and were allocated 12 hours of compute time, but finished well-within that period.

In the following sections, we present the specific methods and results used for each differential equation. All experiments are repeated for 10 different random seeds with means and standard deviations of all results shown in Table 1.

## 3.2 LAPLACE'S EQUATION

Consider a domain given by $\Omega = (0,1) \times (0,1)$, with $\partial\Omega$ corresponding to the boundary lines at $x = 0, y = 0, x = 1, y = 1$.

We construct a target function given by:

$$f(x,y) = \text{Re}\left\{ \frac{1}{(z - 1.2 - 0.5i)(z + 0.2 - 0.5i)(z - 0.5 + 0.2i)(z - 0.5 - 1.2i)} \right\}, \quad (16)$$

where $z = x + iy$. Note that we construct a holomorphic function in Eq. equation 16 that is not holomorphic throughout the entirety of $\mathbb{C}$. This prevents the solution of the function from appearing as a trivial solution of the superposition and holomorphic neural networks.

As boundary conditions, we take Dirichlet conditions of $f(x,y)$ on $\partial\Omega$ for the benchmark that we refer to as Laplace 1, and we take Neumann boundary conditions in the y-direction at $y = 1$ for the case of Laplace 2.

For the superposition networks, we take as activation functions the real parts of the following functions $\sin z$, $\sin z^2$, $\sin^2 z$, $\sin \sin z$, $e^z$, $e^{\sin z}$, $\sin e^z$ and $e^{z^2}$ as our chosen activation functions in Eq. equation 5, with 32 repeats of functions each. We implement elements of the Euclidean symmetry group in Eq. equation 6 as a rotation followed by a translation, with half of them mirrored to allow for orientation to be preserved or flipped. Initial angles of rotations are sampled uniformly on $(0, 2\pi)$. Shift amounts in Eq. equation 6 are initialised to zero. All other parameters in the superposition network initialised to zero. For holomorphic neural networks and PINNs, we use MLPs

with three hidden layers of width 64. We provide training points on 128 evenly spaced points on each of the lines $x = 0, x = 1, y = 0, y = 1$ for collocation points on the boundary, and on 1024 uniformly-random distributed points on the interior of $\Omega$.

We find that neural network architectures constrained to follow the Laplace equation tend to perform better than PINNs in our benchmarks, as demonstrated in table 1. Holomorphic neural networks maintain a strong performance across both benchmarks, surpassing superposition networks. However, holomorphic neural networks are only applicable towards modelling Laplace's equation in 2D,wwhereas the superposition network methodology applies to arbitrary linear differential equations in arbitrary dimensions.

### 3.3 HEAT EQUATION

We consider two heat equation benchmarks in two-spatial dimensions with different initial conditions:

$$\phi(x, y, 0) = \sqrt{e^{-5((x-0.5)^2+(y-0.5)^2)}(\sin^2 5\pi x + \cos^2 3\pi y)} \tag{17}$$

and

$$\phi(x, y, 0) = e^{-5((x-0.5)^2+10(y-0.5)^2)} - e^{-20((x-0.5)^2+5(y-0.7)^2)} - e^{-20((x-0.5)^2+5(y-0.3)^2)} \tag{18}$$

for what we refer to as Heat1 and Heat2 in table 1, respectively.

As for boundary conditions, we take mixed Dirichlet and Neumann boundary conditions such as

$$\phi(0, y, t) = \phi(1, y, t) = 0 \qquad \left.\frac{\partial \phi}{\partial y}\right|_{y=0} = \left.\frac{\partial \phi}{\partial y}\right|_{y=1} = 0 \tag{19}$$

for Heat1 (17) and

$$\left.\frac{\partial \phi}{\partial x}\right|_{x=0} = \left.\frac{\partial \phi}{\partial x}\right|_{x=1} = 0 \qquad \phi(x, 0, t) = \phi(x, 1, t) = 0 \tag{20}$$

for Heat2 (18).

As with the Laplace equation in section 3.2, we are careful to ensure that the initial conditions are not directly representable by a single term of a superposition, thus many superposition bases are required to approximate the resulting solution.

In this instance, we only benchmark against PINNs since NCL and holomorphic neural networks do not represent the heat equation.

For the superposition networks, we implement architectures as per section 2.2.3 with 64 trainable components. We initialise all parameters in Eq. equation 10 and Eq. equation 5 random-uniformly on $(0, 1)$, with the exception of $\theta_{x1i}$ and $\theta_{y1i}$, which are initialised random-uniformly on $(0, 10)$. We use MLPs with 3 hidden layers of width 64 for PINNs.

We find that the superposition network methodology is able to perform better than the PINN on both problems, even though the approximation of the initial solution is not as precise. The diffusion phenomenon is, however, very well captured, proving that the proposed method can handle the extra layer of complexity brought by the time dependency of the differential equation.

### 3.4 INCOMPRESSIBLE NAVIER-STOKES EQUATION

In this section, we demonstrate the challenges remaining in physics-informed optimsation for practical problems. We consider solutions to the incompressible steady-state Navier-Stokes equation:

$$(\mathbf{u} \cdot \nabla)\,\mathbf{u} = \nu\nabla^2\mathbf{u} - \frac{1}{\rho}\nabla\mathbf{p}, \tag{21}$$

with the additional constraint that $\nabla \cdot u = 0$ everywhere. We denote the x and y components of $\mathbf{u}$ with unbolded symbols $u$ and $v$, respectively.

We take $\Omega$ as $(0, 0.5) \times (0, 0.1)$, with two circles of radius 0.05 removed from the rectangle. The centres of the circles lie at $(0.15, 0.1)$ and $(0.35, 0.0)$. As boundary conditions, we take $\mathbf{u} = (0.1, 0)$

at $x = 0$, $\mathbf{u} = w$ on the circles, $p = 0$ at $x = 0.5$, and $v = 0$ elsewhere. The constraint of $v = 0$ corresponds to imposing a reflection of the velocity field parallel to the x-axis at $y = 0.15$ and $y = 0$, thus this represents a tiled structure.

For the superposition network setups, we follow to use the following forms for $\sigma_i$ in Eq. equation 5: $(\cos(x + y), -\cos(x + y)), (e^{x+y}, -e^{x+y}), (x \cos(xy), -y \cos(xy))$. Our initialisation and number of symmetry groups follow the same parameters as per section 3.2. For PINNs, NCL and the networks representing $p$ we use MLPs with three hidden layers of width 32 each. The PINN has an additional loss term for divergence-free constraints. We sample equally spaced points on $\partial\Omega$ such that each line segment has a density of 1000 collocation points per unit length, with the exception of each semi-circle which has 100 collocation points on the semi-circle each. The interior collocation points are formed with rejection sampling with 1024 random uniformly distributed points distributed over the rectangle $(0, 0.5) \times (0, 0.1)$.

We find it noteworthy that in this scenario where nonlinear Navier-Stokes constraints cannot be embedded within the architecture, convergence is problematic for all methods.

### 3.5 BURGERS' EQUATION

We construct a benchmark defined as per the 1D-Burgers' equation in equation 11 with $\nu = 0.1$, initial conditions of $u(x, 0) = \exp(-50(x - 0.6)^2) - \exp(-50(x - 0.4)^2)$ and boundary conditions of $u(0, t) = 0$ and $u(1, t) = 1$, with $\Omega = (0, 1)^2$.

For superposition networks, we consider 100 components with parameters chosen analogously to the heat equation experiments of section 3.3. We use 64 evenly spaced points on each Dirichlet boundary for imposing boundary conditions and 1024 collocation points on the interior for PINN based methods.

## 4 DISCUSSION

Motivated by the difficulty of training physics-informed regularisation, this work has sought to embed linear differential equation constraints within neural network architectures, and also theoretically demonstrated the possibility of extending this to some non-linear differential equations. We suggested a systematic approach to further extend our approach to include additional differential operators, and hence ODE/PDEs, as well as illustrating ad-hoc constructions for notable cases. Crucially, ensuring convergence guarantees for equations at the interior of the domain might enable the adoption of neural architectures in more critical applications than is currently targeted. Numerical investigations support a perspective that favours embedding differential equations constraints via architectural design rather than regularisation. However, we emphasise some limitations in our approaches. While we present Lie-group symmetries and manufactured solutions of certain linear-differential equations, we do not present methical ways to derive such solutions for arbitrary linear differential equations. In practice, for other diffrential equations, a practitioner might employ techniques such as a trial ansatz, using physically-motivated arguments, or attempting standard approaches used in nonlinear differential equations (Hydon, 2000). We hope that our work motivates further research into embedding differential equation constraints directly into neural network architectures.

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
