# OpenReview forum: "Neural Superposition Networks"
_ICLR.cc/2025/Conference — Submitted to ICLR 2025_

### Official Review · Reviewer_v1ki · 2024-10-30

**Soundness:** 3
**Presentation:** 2
**Contribution:** 2
**Rating:** 3
**Confidence:** 3

**Summary:**

This work reinterprets the existing methods of using regularisation or architecture design as linear superpositions of general formulated solutions and uses this notion of superposition to develop architectures that satisfy various differential equations. Besides new architectures for Laplace’s equation and divergence-free fields, the work proposes constraints suitable for the heat equation and some nonlinear differential equations. Empirical results on solving some toy example PDEs show better performances than existing methods such as PINN, showing that embedding differential equation constraints directly into neural network architectures can improve performance.

**Strengths:**

1. The paper shows great understanding of the concept of physics-inspired/constrained machine learning, and proposed a novel approach to solve physical problem ground up.
2. The proposed method is mathematically sound, and the experiments presented are intuitive.

**Weaknesses:**

1. The presentation of the paper is not clear. It would be much more clearer if the paper is structured in a way that the intention of solving physical problems (e.g. PDEs) is clear at the beginning.
2. The empirical results are weak, and only toy example level.

**Questions:**

Whereas existing methods like PINN does not scale to high dimensional PDEs, how is the scaling property of the proposed work?

---

> ### Author Response · Authors · 2024-11-21
>
> We’re grateful for the reviewer’s input. We will emphasise the following points in our next revision that we feel the reviewer had missed:
>
> 1. Embedding differential constraints within architectures leads to more reliable convergence, as opposed to when such constraints are imposed via regularisation.
> 2. Existing architectures with differential constraints [1,2] have interpretations as superposition networks.
> 3. We present differentially constrained architectures adhering to the Burgers equation and Heat equation for the first time. This work adds to existing literature [1,2] previously applying only to divergence-free fields and Laplace’s equation.
>
> ## Response to Weaknesses
>
> 1. We will emphasise that solving differential equations by constrained architectures is the main problem setting in the first paragraph of the introduction in the next version.
> 2. Though the empirical results are simple, we find it noteworthy that many published regularisation-based approaches do not converge well on them. Thus, we judge them to be of a suitable level for benchmarks.
>
> ## Response to Questions
>
> 1. We believe that we should have some scaling advantages over PINNs. As opposed to optimising on interiors of domains as PINNs require, our methods only require optimisation of the boundary. Since the boundary is of a lower-dimensionality than the interior, then we naturally cast problems into a lower dimensional setting. Furthermore, by not requiring higher-order derivatives to impose physics-informed regularisation on boundary interiors, we circumvent a potentially computationally-expensive aspect of PINNs.

---

> > ### Comment · Reviewer_v1ki · 2024-11-24
> >
> > I thank the author for the response. However, as in general the paper is unclear (as indicated by the fact that all reviewers have missed at least some points the authors were trying to convey), I believe the paper can benefit from another round of submission. I will keep my score at reject.

---

### Official Review · Reviewer_x8QK · 2024-11-04

**Soundness:** 2
**Presentation:** 2
**Contribution:** 2
**Rating:** 3
**Confidence:** 4

**Summary:**

This paper presents a unifying framework Machine learning-based solving of partial differential equations, where the differential constraints themselves are embedded into the architecture of the model. By introducing certain linear superpositions of such constrained mappings, this contribution presents new architectures capable of solving a number of PDEs with strong enforcement of the differential constraints

**Strengths:**

- The contribution introduces a novel architecture based on a superposition of ML-based solutions satisfying certain differential constraints
- A benchmark for a number of linear and nonlinear partial differential equations is establshed

**Weaknesses:**

- While the superposition NN seems to give better results than PINNs, which impose the constraint weakly through the loss, I fail to see why one would use a superposition NN against an architecture that directly imposes the constraint; in the case of the holomorphic NN for Laplace and the NCL for Navier-Stokes, those simple architectures seem to give better results than the superposition NN.
- For the Heat Equation and Burgers, no test is offered against an architecture solely enforcing the constraint, without the superposition
- It is unclear to me  whether superposition really provides any edge in the case of nonlinear PDEs. The Cole-Hopf transform works indeed for Burgers but this is a particular case, and for Navier-Stokes it appears none of the models manage to achieve good convergence.
- Regarding Navier-Stokes, could the authors identify why none of the methods succeed in achieving convergence ? ML methods (Unets, FNOs) were shown to work in the past for solving steady state incompressible NS, even on more complex domains than the one suggested
- Certain notations are a bit confusing: eq 9, the right hand side does not depend on i
- Certain typos: avoid repeating "Eq. equation", naybe just "Eq" or "equation"

**Questions:**

- Could the authors test a non-superposition NN which strongly enforces the heat equation constraint and compare against the superposition results for the heat equation (and Burgers ?)
- For Navier-Stokes, it would be relevant to study a case where at least some of the solvers proposed in the benchmark do converge, whether this means adding solvers to the benchmark, or reducing the difficulty of the problem at hand (maybe considering a simple square with periodic boundary conditions)

---

> ### Author Response · Authors · 2024-11-21
>
> # **Response to Reviewer x8QK**
>
> We’re grateful for the reviewer’s input. A lot of their points and questions speak to miscommunications on our side in the initial version. The key points we wish to convey in the paper are:
>
> 1. Embedding differential constraints within architectures leads to more reliable convergence, as opposed to when such constraints are imposed via regularisation.
> 2. Existing architectures with differential constraints [1,2] have interpretations as superposition networks.
> 3. We present differentially constrained architectures adhering to the Burgers equation and Heat equation for the first time. This work adds to existing literature [1,2] previously applying only to divergence-free fields and Laplace’s equation.
>
> We will emphasise these points as our core contributions in the next version. We feel these interesting points for the field, especially since regularisation is currently the predominant form of imposing differential constraints into neural networks.
>
> ## Response to Weaknesses
>
> 1. One of our main points is to motivate the use of differential constraints within architectures where possible and agree that using holomorphic/NCL approaches make sense where applicable, namely the 2D Laplace equation and divergence free fields. Using architectural constraints is currently uncommon praxis, and we wish to motivate its use more. Our methods are more general than holomorphic neural networks/NCL and apply to situations where there are currently no published architectures available.
> 2. As far as we are aware, our work is the first to develop architectures constrained to the heat and Burgers’ equations. This is why we do not include alternative architecturally-constrained approaches for these benchmarks.
> 3. It is true that performance for Navier-Stokes is poor across all approaches. However, it is also the case that none of the approaches is able to constrain to solutions of Navier-Stokes architecturally. We might interpret the poor performance of the Navier-Stokes approaches reinforces our principal point that including architectural constraints where possible is beneficial for convergence.
> 4. We believe the Navier-Stokes equations are hard to solve in an unsupervised physics-informed setting. The methods outlined by the reviewer, e.g. Fourier-Neural Operator, are learning solutions in a data-driven supervised-learning manner. In constrast, all the approaches we benchmark are data-independent and unsupervised. Thus, we see the strong performance of FNOs in other work as belonging to a very different problem setting.
> 5. We are grateful for the reviewer’s feedback on presentation improvements, and these will be addressed in the next revision.
>
> ## Response to Questions
>
> 1. To our knowledge, there are no alternative architectures that satisfy the heat and Burgers’ equations and we are the first to present such architectures. This is why we did not include these in any benchmarks. We did include architecturally-constrained architectures where they exist (i.e. NCL and Holomorphic). A part of our work is that we wish to motivate the development of more architecturally-constrained architectures given that we believe they perform favourably compared to the predominant approach of physics-informed regularisation.
> 2. Our Navier-Stokes benchmark is already quite simple. In the absence of obstructions in the domain, the solution would be a constant velocity field. And we also consider steady-states instead of transient solutions. To some extent, our results reinforce the challenges facing phsyics-informed machine learning.

---

> > ### Comment · Reviewer_x8QK · 2024-11-25
> >
> > I thank the authors for their answers. In particular, the use of superposition networks for the Heat Equation is clearer now. But indeed, a study such as the one provided in your answer to Reviewer DiTu, with the impact of increasing number of superposition terms, is of paramount importance for the paper and would help justify in practice the use of these networks.

---

> > > ### Author Response · Authors · 2024-12-02
> > >
> > > Similarly to what we did on the Laplace equation, we performed a study on the Heat equation with Dirichlet boundary conditions (i.e. heat1) to show that increasing the number of components per basis function has again a direct and noticeable impact on the quality of the results.
> > >
> > > | __Num Transformations__ | __mean RMSE +/- std(RMSE)__ |
> > > | --- | --- |
> > > | 4 | 3.94e-2 +/- 1.3e-2 |
> > > | 8 | 2.69e-2 +/- 4e-3 |
> > > | 16 | 1.95e-2 +/- 8.6e-3 |
> > > | 32 | 9.11e-3 +/- 2.71e-3 |
> > > | 64 | 7.97e-3 +/- 7.3e-05 |

---

### Official Review · Reviewer_DiTu · 2024-11-04

**Soundness:** 3
**Presentation:** 2
**Contribution:** 3
**Rating:** 6
**Confidence:** 4

**Summary:**

This paper introduces a framework for constructing neural networks that inherently satisfy linear differential equations through the principle of superposition. The authors show how their framework unifies previous architectures for divergence-free fields and Laplace's equation, while also enabling novel architectures for the heat equation and even some nonlinear equations like Burgers' equation. The work includes theoretical development and empirical validation across multiple differential equations.

**Strengths:**

1. The theoretical foundation is strong, with clear mathematical development based on superposition principles. The framework elegantly unifies existing architectures while providing rigorous justification for new ones.
2. The practical applicability is impressive, demonstrating success across multiple important differential equations. The method works well for Laplace's equation, heat equation, and even extends to some nonlinear cases like Burgers' equation through clever transformations.
3. The empirical evaluation is comprehensive, comparing against multiple baselines including physics-informed neural networks and specialized architectures. Results are presented with proper statistical analysis across multiple random seeds and clear performance metrics.
4. The method successfully generates novel architectures that outperform existing approaches. For example, their heat equation architecture shows better performance than PINNs while requiring fewer iterations, and their treatment of Burgers' equation is particularly innovative.

**Weaknesses:**

1. The theoretical guarantees are limited. While the paper proves that solutions satisfy the differential equations, there are no guarantees about approximation capabilities or convergence. The lack of universality proofs significantly limits our understanding of what the networks can and cannot represent.
2. The approach primarily targets linear differential equations, with extensions to nonlinear equations feeling somewhat ad hoc. The Cole-Hopf transformation for Burgers' equation, while clever, doesn't suggest a general strategy for handling nonlinear equations.
3. Implementation details crucial for reproduction are scattered throughout the paper and supplementary materials. The handling of boundary conditions and parameter initialization strategies needs more thorough treatment in the main text.
4. The experimental evaluation, while thorough for the chosen problems, focuses on relatively simple test cases. There's limited exploration of higher-dimensional problems or complex geometries that would be encountered in real applications.
5. The computational complexity analysis is insufficient. While the method shows good performance, there's no detailed analysis of memory requirements, training stability, or computation time compared to alternatives.

**Questions:**

1. What is the relationship between network size and approximation capability? Can you characterize the class of functions that can be represented as the number of superposition terms increases?
2. How does the method handle mixed boundary conditions in practice? The paper shows some examples, but what are the limitations when dealing with complex boundary geometries or discontinuous boundary conditions?
3. Can you develop a more systematic approach to handling nonlinear equations beyond the specific case of Burgers' equation? What properties must a nonlinear equation have for your approach to be applicable?
4. How does the performance scale with dimensionality? While 2D examples are shown, many practical applications require solving 3D or higher-dimensional problems.
5. The Lie group symmetries play a crucial role in your construction. How can one systematically identify and incorporate appropriate symmetries for new differential equations?
6. The initialization of network parameters seems important for performance. How sensitive is the method to different initialization strategies, and can you provide theoretical guidance for choosing good initializations?
7. For the heat equation results, how well does the method handle different diffusion coefficients? Is there a range of coefficients where the method performs particularly well or poorly?
8. The paper mentions potential extensions to stochastic differential equations. What modifications would be necessary to handle stochastic terms while maintaining the superposition property?

---

> ### Author Response · Authors · 2024-11-21
>
> We’re grateful for the reviewer’s feedback and general appreciation. We’d like to emphasize that the main messages we want to convey with the paper are:
>
> 1. Embedding differential constraints within architectures leads to more reliable convergence, as opposed to when such constraints are imposed via regularization.
> 2. Existing architectures with differential constraints [1,2] have interpretations as superposition networks.
> 3. We present differentially constrained architectures adhering to the Burgers equation and Heat equation for the first time. This work adds to existing literature [1,2] previously applying only to divergence-free fields and Laplace’s equation.
>
> ## Response to Weaknesses
>
> 1. Although we provide no guarantee about approximation capabilities or convergence, this situation is shared with other architecturally constrained methods too [1,2]. Universality of differentially-constrained  architectures in general is a topic that has not seen much (or any?) development. It would be of much interest for future work.
> 2. We currently don’t have a general method to apply superposition networks to non-linear differential equations, but the example provided for Burgers equation can is a first step.
> 3. We will reorganize and improve explanation of such details in the updated version of the paper to improve readability and clarity.
> 4. Our benchmarks, though simple, do highlight that some existing and popular PINN approaches fail to converge suitably, thus we judge them to be of a suitable level to judge the current state of physics-informed machine learning.
> 5. Since superposition networks only need to be trained on boundaries, the computational cost is generally less lower than PINNs, since no evaluation on internal collocation points is needed during training. Furthermore, since we avoid the need to calculate higher order derivatives on collocation points on the interior of the domain, we anticipate reduced memory requirements.
>
> ## Response to Questions
>
> 1. We performed a study on Laplace equation with Dirichlet boundary conditions to show that increasing the number of harmonics and the number of components per basis function has a direct impact on the quality of the results
>
>  | Number of Harmonics | Num Transformations per Harmonic |mean  RMSE +/- std(RMSE)|
> |----|----|----|
> |2|8|0.03 +/- 0.02|
> |2|32|0.015 +/- 0.007|
> |4|8|0.008 +/- 0.005|
> |4|32|0.007 +/- 0.003|
>
> 2. In principle handling mixed boundary conditions would be done as with PINNs. Our Laplace2 benchmark does consider a combination of Neumann and Dirichlet boundaries and demonstrates good performance of superposition networks.
> 3.  In general, for our approach one needs to find a suitable transformation that allows mapping between a nonlinear and linear problem. We acknowledge that this is a manual method. Previous papers similarly satisfy specific differential equations with specific tricks [1,2]. In the absence of a general method, we wish to motivate further development of specific architectures, even if they require tricks.
> 4. Wee do find that many simple 2D problems already prove challenging for PINNs and provide a worthy set of benchmarks. We hope there are favourable aspects to superposition networks in regards to scaling up: since they optimise only on boundaries, they are always solving an optimisation procedure in a lower dimensionality than the corresponding PINN problem.
> 5. There are some methodical procedures to be tried which we point readers to the  [3, 4]. In practice, there are also alternative paths as well, e.g. hypothesising an ansatz and substituting into the original differential equation, or by physical reasoning (e.g. rotation and translation invariance should apply to many physical problems).
> 6. The initialization technique is the one reported in sections 3.2, 3.4 and 3.5. In general it stands to reason that the best initialisation method would be problem-dependent and perhaps chosen via established hyperparameter optimisation methods.
> 7. We did not test multiple values for the diffusion coefficient in the heat equation. However, since the heat equation is also solved under-the-hood in the Burgers setup with different coefficients, we do tacitly do test two separate diffusion coefficients.
> 8. In general the derivation of sample paths does not follow from superposition. However, the probability distribution in time of several SDEs might be calculated via superposition solutions of the associated Fokker-Planck equation, which describes the time-dependent probability distribution of an SDE in terms of a linear equation in the probability dis tribution.
>
> [1] Richter-Powell et al, Neural Conservation Laws: A Divergence-Free Perspective, NeurIPS 2022
>
> [2] Ghosh et al, Harmonic Neural Networks, ICML 2024
>
> [3] Hydon, P. E.  Symmetry methods for differential equations. Cambridge University Press, 2000
>
> [4] Gray, R. J. How to calculate all point symmetries of linear and linearizable differential equations, Proc Roy Soc A, 2015

---

> > ### Comment · Reviewer_DiTu · 2024-11-26
> >
> > Thank you for your detailed and thoughtful response. I appreciate the clarifications on the scaling advantages, the role of superposition terms, and your acknowledgment of the limitations regarding universality and nonlinear PDEs. While your responses address some concerns and provide valuable insights, I believe the empirical and theoretical gaps identified earlier still warrant further development. I will maintain my score.

---

### Official Review · Reviewer_j3C5 · 2024-11-05

**Soundness:** 2
**Presentation:** 2
**Contribution:** 2
**Rating:** 3
**Confidence:** 2

**Summary:**

This submission proposes a method for neural networks to satisfy linear differential equations with boundary conditions of the form given in equation (1). The core idea is to linearly combine neural networks that satisfy the differential equation on an open set Omega, and optimize the parameters so that the boundary conditions are met. This method generalizes previous methods developed for particular differential equations. This technique is applied to six types of equations: four linear (Laplace 1, Laplace 2, Heat 1, and Heat 2), and two nonlinear (Navier-Stokes and Burgers'), and compared with other approaches such as PINNs, RAR, AA, and Holomorphic.

**Strengths:**

- The core idea of the method is simple to understand and well-illustrated in Figure 1.
- The submission does not over-claim the advantages of the method, and limitations are discussed (l. 437, l. 526).
- Experimental results are carefully reported with error margins in Table 1.
- Details are given on the different forms of the superpositions networks considered (2.2.(1-4), 3.(2-5))
- Code to reproduce the experiments is provided as supplementary material.

**Weaknesses:**

I would like to emphasize that I am not an expert in PINNs and differential equation solving using neural networks. Nevertheless, I find that the contributions of the submission are not strong enough to justify publication at ICLR, primarily due to the following reasons:

* While presented as a methodical contribution, the idea of a superposition network seems quite straightforward.
* Your method outperforms other benchmarks only for the Heat 2 equation. The results for the Burgers equation are not statistically significant, and for all other benchmarks, your method is outperformed by another. While I appreciate that there is no overclaim in the paper regarding the advantages of the method, I would like to understand its true practical advantage.
* There are no theoretical results (theorem/proposition/lemma) in the submission. While this could be acceptable with strong experimental support, the current results are not convincing (see above).

In addition to these concerns, I find the paper to suffer from presentation issues:
* The paper lacks figures to illustrate the method beyond a single schematic (Figure 1).
*  Several parts of the submission are in my opinion difficult to understand:
    * More details should be provided upfront about the holomorphic baseline method (l. 161) instead of later in the text (l. 356).
    * The Lie group discussion around l.197 is hard to follow and needs additional explanations.
    * The remark on l. 240 is hard to parse. For instance the second and third sentences sounds like the same.
    * The indices used for θ in equation 10 are not defined.
    * The explanations of the group action are insufficient (around l. 201 and l. 224).
    * Lines 330-331 are unclear and seem out of place.
    * The meaning of "extra weights" in lines 339-341 is not clear.
* Many details, such as the different forms of superposition networks (2.2.(1-4), 3.(2-5)), could be moved to the appendix to allow more space for explaining complex concepts like the group action and Lie group aspects.
* Minor Issues: There are typos like "Eq. Equation" throughout the paper (e.g., l. 87).

**Questions:**

* Why not apply your method to other differential equations than the ones presented here?
* Is there any computational advantage to your method? Since you only optimize on the boundary, do you observe faster convergence to the solution?
* Could you clarify the meaning of lines 330-331 ?
* What do you mean by "extra weights" in lines 339-341? Is it a scalar multiplicative factor in front of one of the two terms?

---

> ### Author Response · Authors · 2024-11-21
>
> We’re grateful for the reviewer’s time and feedback. In particular, given their review, we note that our initial version was missing clarity on our key overall messages, which are:
>
> 1. Embedding differential constraints within architectures leads to more reliable convergence, as opposed to when such constraints are imposed via regularisation.
> 2. Existing architectures with differential constraints [1,2] have interpretations as superposition networks.
> 3. We present differentially constrained architectures adhering to the Burgers equation and Heat equation for the first time. This work adds to existing literature [1,2] previously applying only to divergence-free fields and Laplace’s equation.
>
> We will emphasise these points as our core contributions in the next version. We feel these interesting points for the field, especially since regularisation is currently the predominant form of imposing differential constraints into neural networks. More details below.
>
> ## Response to Weaknesses
>
> 1. *On the straightforwardness of methodology*. We agree that superposition networks might be seen as straightforward. However, we believe they are novel, applicable and of wider interest. Superposition provides a unifying perspective on previous neural networks architectures [1,2] and also a prism through which to construct new ones.
> 2. *On the practical advantage demonstrated empirically.* Our results support that embedding differential constraints within architectures (as opposed to via regularisation) is beneficial to reliable convergence. The strong performance of holomorphic neural networks reinforces this point. We hope this work can motivate research towards embedding novel differential constraints in architectures, a topic which has been less explored than regularisation-based approaches. We will segregate our results table into architectures which impose differential constraints and those which do not in our rebuttal version.
> 3. *On theoretical contributions.* We agree that we do not have theoretical guarantees on performance. **Our principal theoretical contributions are i) that superposition provides a unifying perspective of some recently published architectures [1,2] and ii) the guaranteed adherence of superposition networks to linear differential equation constraints, along with discussions on the relevance of domain topology to universality.
> 4. *Presentation improvements.* We are grateful for the reviewer for their suggestions. Instead of introducing an extra figure outlining the method (which we hope is covered by the current Figure 1) we will introduce a motivational example section at the start of the theory section to provide an accessible and fast overview of core ideas of superposition and Lie groups applied to the Laplace equation for the interested reader. We hope this makes the principle points of the methodology much clearer. We agree on the remaining points of the reviewer and provide corrections in the new revision.
>
> ## Response to Questions
>
> 1. We focus on a subset of common and widely applicable PDEs. We note that previous literature [1,2] focussed on single PDEs. Our exposition includes those specific PDEs as well as novel PDEs on top of that. From this perspective, one might view our work as more general than those existing works.
> 2. There are potential computational benefits indeed from only optimising on the boundary. The boundary is of a lower dimension than the interior, so optimising on it is less computationally expensive. Also, taking higher-order derivatives on the interior is circumvented. We will include timings for each method in the next version. Though it is difficult to draw fair comparisons of timings for fundamentally different optimisation problems (optimising on the interior vs optimising on the boundary) and different architectures (what size multilayer perceptron should correspond to what size superposition network?).
> 3. For the training of PINNs, we require distributions of collocation points on boundaries and interiors of domains over which we optimise our losses on. Lines 330-331 introduce probability distributions over these points so that we can succinctly write the resulting losses in terms of expectations over these distributions in the resulting sections. We will clarify this in the rebuttal version.
> 4. By extra weights, we mean that we introduce coefficients which multiply the loss of the interior/boundary terms in Eq 14 to give them extra importance. This will be clarified in the next version.
>
> [1] Richter-Powell et al, Neural Conservation Laws: A Divergence-Free Perspective, NeurIPS 2022
>
> [2] Ghosh et al, Harmonic Neural Networks, ICML 2024

---

> ### Comment · Reviewer_j3C5 · 2024-11-22
>
> Thank you for your response.
>
> > There are potential computational benefits indeed from only optimising on the boundary. The boundary is of a lower dimension than the interior, so optimising on it is less computationally expensive. Also, taking higher-order derivatives on the interior is circumvented. We will include timings for each method in the next version. Though it is difficult
>
> Why is it difficult ? Could you provide running times for one of the PDEs in the paper? I would be interested in seeing whether there actually is an improvement. Thank you

---

> > ### Author Response · Authors · 2024-11-22
> >
> > We apologise to the reviewer. The sentence was incomplete. It was meant to read "Though it is difficult to draw fair comparisons of timings for fundamentally different optimisation problems (optimising on the interior vs optimising on the boundary) and different architectures (what size multilayer perceptron should correspond to what size superposition network?).
> >
> > The timings themselves are not difficult to add, and will be included in the next version.

---

> > > ### Comment · Reviewer_j3C5 · 2024-11-22
> > >
> > > Thanks for the clarification.
> > >
> > > If you have time to provide me before the end of the discussion period with running times for some of the PDEs I would be interested.

---

### Meta-Review · Area_Chair_5C2d · 2024-12-20

**Metareview:**

This paper proposes a method for building neural networks that satisfy linear differential equations. The idea is to linearly combine neural networks that satisfy the differential equation and optimize the parameters to meet the boundary conditions. This technique is applied to some basic linear and nonlinear differential equations and compared with other baseline approaches.

The reviewers appreciate the mathematical soundness of the method and the intuitiveness of the presented experiments. They also value the authors' provision of the code to reproduce the experiments and the fact that the paper does not overstate the method's advantages. Furthermore, they commend the work's theoretical foundation, believing that the proposed framework elegantly unifies existing architectures while providing rigorous justification for new ones. The reviewers also highlight the successful generation of novel architectures that outperform existing approaches.

However, the reviewers also raised multiple questions and concerns. Reviewers j3C5 and DiTu share similar concerns, believing the paper currently falls short of being categorized as a strong work in either theoretical or empirical directions. While it clearly lacks any theoretical analysis, the current level of experimentation is not sufficient to treat it as a solid empirical work either. They also both wonder about the computational cost of the method. The authors provided a response, and while the reviewers appreciate that the response addressed some of their concerns, they did not completely resolve them. Reviewer x8QK asks several questions: why one would use a superposition NN against an architecture that directly imposes the desired constraint; why no test was offered for the heat equation against an architecture solely enforcing the constraint without the superposition; and how to explain the meaningful gains by the superposition networks in the nonlinear PDE cases. The authors' responses helped clarify some of these issues. However, the reviewer believes a study proposed by the authors themselves to Reviewer DiTu, examining the impact of increasing the number of superposition terms, would be needed to justify the use of these networks in practice.

While I agree with the reviewers that the paper proposes an elegant approach to generate architectures for solving PDEs, I also concur that it could be made much stronger by addressing the unresolved concerns.

**Additional Comments On Reviewer Discussion:**

Reviewers j3C5 and DiTu share similar concerns, believing the paper currently falls short of being categorized as a strong work in either theoretical or empirical directions. While it clearly lacks any theoretical analysis, the current level of experimentation is not sufficient to treat it as a solid empirical work either. They also both wonder about the computational cost of the method. The authors provided a response, and while the reviewers appreciate that the response addressed some of their concerns, they did not completely resolve them. Reviewer x8QK asks several questions: why one would use a superposition NN against an architecture that directly imposes the desired constraint; why no test was offered for the heat equation against an architecture solely enforcing the constraint without the superposition; and how to explain the meaningful gains by the superposition networks in the nonlinear PDE cases. The authors' responses helped clarify some of these issues. However, the reviewer believes a study proposed by the authors themselves to Reviewer DiTu, examining the impact of increasing the number of superposition terms, would be needed to justify the use of these networks in practice.

---

### Decision · Program_Chairs · 2025-01-22

Reject